

# HighResMIP versions of EC-Earth: EC-Earth3P and EC-Earth3P-HR. Description, model performance, data handling and validation

Rein Haarsma[1], Mario Acosta[5], Rena Bakhshi[2], Pierre-Antoine Bretonnière[5], Louis-Philippe Caron[5], Miguel Castrillo[5], Susanna Corti[4], Paolo Davini[4], Eleftheria Exarchou[5], Federico Fabiano[4], Uwe Fladrich[3], Ramon Fuentes Franco[3], Javier García-Serrano[6,5], Jost von Hardenberg[4], Torben Koenigk[3], Xavier Levine[5], Virna Meccia[4], Twan van Noije[1], Gijs van den Oord[2], Froila M. Palmeiro[6], Mario Rodrigo[6], Yohan Ruprich-Robert[5], Philippe Le
Sager[1], Etienne Tourigny[5], Shiyu Wang[3], Michiel van Weele[1], Klaus Wyser[3].

1. Royal Netherlands Meteorological Institute (KNMI), De Bilt, Netherland
2. Netherlands eScience Center, Amsterdam, Netherlands
3. Swedish Meteorological and Hydrological Institute (SMHI), Norrköping, Sweden
4. Institute of Atmospheric Sciences and Climate, Consiglio Nazionale delle Ricerche (ISAC-CNR), Italy
5. Barcelona Supercomputing Center (BSC), Barcelona, Spain
6. Group of Meteorology, Universitat de Barcelona (UB), Barcelona, Spain

*Correspondence to:* Rein Haarsma (rein.haarsma@knmi.nl)



**Abstract** A new global high-resolution coupled climate model, EC-Earth3P-HR has been developed by the EC-Earth consortium, with a resolution of approximately 40 km for the atmosphere and 0.25 degree for the ocean, alongside with a standard resolution version of the model, EC-Earth3P (80 km atmosphere, 1.0 degree ocean). The model forcing and simulations follow the HighResMIP protocol. According to this protocol all simulations are made with both high and standard resolutions. The model has been optimized with respect to scalability, performance, data-storage and post-processing. In accordance with the HighResMIP protocol no specific tuning for the high resolution version has been applied.

Increasing horizontal resolution does not result in a general reduction of biases and overall improvement of the variability, and deteriorating impacts can be detected for specific regions and phenomena such as some Euro-Atlantic weather regimes, whereas others such as El Niño-Southern Oscillation show a clear improvement in their spatial structure. The omission of specific tuning might be responsible for this.

The shortness of the spin-up, as prescribed by the HighResMIP protocol, prevented the model to reach equilibrium. The trend in the control and historical simulations, however, appeared to be similar, resulting in a warming trend, obtained by subtracting the control from the historical simulation, close to the observational one.



## 1 Introduction

Recent studies with global high-resolution climate models have demonstrated the added value of enhanced
horizontal atmospheric and oceanic resolution compared to the output from models in the coupled model
intercomparison project phase 3 and 5 (CMIP3 and CMIP5) archive. An overview and discussion of those studies
has been given in Haarsma et al. (2016) and Roberts et al. (2018). Coordinated global high-resolution experiments
were, however, lacking, which induced the launch of the CMIP6 endorsed High Resolution Model
Intercomparison Project (HighResMIP). The protocol of HighResMIP is described in detail in Haarsma et al.
(2016). Due to the large computational cost that high horizontal resolution implies, the time period for simulations
in the HighResMIP protocol ranges from 1950 to 2050. The minimal required atmospheric and oceanic resolution
for HighResMIP is about 50 km and 0.25⁰ respectively.

EC-Earth is a global coupled climate model (Hazeleger et al., 2010, 2012) that has been developed by a consortium
of European institutes consisting to this day of 27 research institutes. Simulations with EC-Earth2 contributed to
the CMIP5 archive, and numerous studies performed with the EC-Earth model appeared in peer-reviewed
literature and contributed to the fifth assessment report (AR5) of the IPCC (Intergovernmental Panel on Climate
Change) (IPCC, 2013). EC-Earth is used in a wide range of studies from paleo-research to climate projections,
including also seasonal (Bellprat et al. 2016; Prodhomme et al., 2016; Haarsma et al., 2019) and decadal forecasts
(Guemas et al., 2013, 2015; Doblas-Reyes et al., 2013; Caron et al., 2014, Soraju-Morali et al., 2019, Koenigk et
al., 2013, Koenigk and Brodeau, 2014, Brodeau and Koenigk, 2016).

In preparation for CMIP6, a new version of EC-Earth, namely EC-Earth3, has been developed (Doescher et al.,
2019). This has been used for the DECK  (Diagnostic, Evaluation and Characterization of Klima) simulations
(Eyring et al., 2016) and several CMIP6-endorsed MIPs. The standard resolution of EC-Earth3 is T255 (~80 km)
for the atmosphere and 1.0⁰ for the ocean, which is too coarse to contribute to HighResMIP. A higher resolution
version of EC-Earth3, therefore, had to be developed. In addition, the HighResMIP protocol demands simplified
aerosol and land schemes (Haarsma et al., 2016).

In section 2, we will describe the HighResMIP version of EC-Earth3 which has been developed within the
European Horizon2020 project PRIMAVERA  (Roberts et al., 2019). For a detailed description of the standard
CMIP6 version of EC-Earth3 and its technical and scientific performances, we refer to Doescher et al. (2019).
High-resolution modeling requires special efforts on scaling, optimization and model performance, which will be
discussed in section 3. In section 3 we also discuss the huge amount of data that is produced by a high-resolution
climate model and requires an efficient post-processing and storage workflow. A summary of the model results
will be given in section 4. In that section we also discuss the issue that for a high resolution coupled simulation it
is not possible to produce a completely spun up state that has reached equilibrium due to limited computer
resources. As a result, the HighResMIP protocol prescribes that the simulations start from an observed initial state.
The drift due to an imbalance of the initial state is then accounted for by performing a control run with constant
forcing alongside the transient run.





## 2 Model description

The model used for HighResMIP is part of the EC-Earth3 family. EC-Earth3 is the successor of EC-Earth2 that
was developed for CMIP5 (Hazeleger et al., 2010, 2012; Sterl et al., 2012). Early versions of EC-Earth3 have
been used by e.g. Batté et al. (2015), Davini et al. (2015) and Koenigk and Brodeau (2017). The versions
developed for HighResMIP are EC-Earth3P for standard resolution and EC-Earth3P-HR for high resolution and
will henceforth be referred to as EC-Earth3P(-HR), respectively. In addition, a very high resolution version EC-
Earth3P-VHR (T1279 (~15 km) atmosphere and 0.12 degree ocean) has been developed and simulations following
the HighResMIP protocol are presently being performed, but not yet available. Compared to EC-Earth2, EC-
Earth3P(-HR) include updated versions of its atmospheric and oceanic model components, as well as a higher
horizontal and vertical resolution in the atmosphere.

The atmospheric component of EC-Earth is the Integrated Forecasting System (IFS) model of the European Centre
for Medium-Range Weather Forecasts (ECMWF). Based on cycle 36r4 of IFS, it is used at T255 and T511 spectral
resolution for EC-Earth3P and EC-Earth3P-HR, respectively. The spectral resolution refers to the highest retained
wavenumber in linear triangular truncation. The spectral grid is combined with a reduced Gaussian grid where the
nonlinear terms and the physics are computed, with a resolution of N128 for EC-Earth3P, N256 for EC-Earth3-
HR and N640 for EC-Earth3P-VHR. The nominal atmospheric resolution is 100 km for EC-Earth3P and 50 km
for EC-Earth3P-HR. Because of the reduced Gaussian grid the grid box distance is not continuous, with a mean
value of 107 km for EC-Earth3P and 54.2 km for EC-Earth3P-HR (Klaver et al., 2019). The number of vertical
levels is 91, vertically resolving the middle atmosphere up to 0.1 hPa. The H-TESSEL model is used for the land
surface (Balsamo et al., 2019) and is an integral part of IFS: for more details see Hazeleger et al. (2012).

The ocean component is the Nucleus for European Modelling of the Ocean (NEMO; Madec, 2008). It uses a tri-
polar grid with poles over northern North America, Siberia and Antarctica and has 75 vertical levels (compared
to 42 levels in the CMIP5 model version and standard EC-Earth3). The so-called ORCA1 configuration (with a
horizontal resolution of about 1 degree) is used in EC-Earth3P whereas the ORCA025 (resolution of about 0.25
degree) is used in EC-Earth3P-HR. The ocean model version is based on NEMO version 3.6 and includes the
Louvain-la-Neuve sea-ice model version 3 (LIM3; Vancoppenolle et al., 2012), which is a dynamic-
thermodynamic sea-ice model with five ice thickness categories. The atmosphere/land and ocean/sea-ice
components are coupled through the OASIS (Ocean, Atmosphere, Sea Ice, Soil) coupler (Valcke and Morel, 2006;
Craig et al., 2017).

The NEMO configuration is based on a set-up developed by the ShaCoNEMO initiative lead by Institute Pierre
Simon Laplace (IPSL) and adapted to the specific atmosphere coupling used in EC-Earth. The remapping of runoff
from the atmospheric grid points to runoff areas on the ocean grid has been re-implemented to be independent of
the grid resolution. This was done by introducing an auxiliary model component and relying on the interpolation
routines provided by the OASIS coupler. In a similar manner, forcing data for atmosphere-only simulations are
passed through a separate model component, which allows to use the same SST and sea-ice forcing data set for
different EC-Earth configurations.



The CMIP6 protocol requests modeling groups to use specific forcing datasets that are common for all participating models. Table 1 lists the forcings that have been implemented in EC-Earth3P(-HR). Because of the HighResMIP protocol, EC-Earth3P(-HR) distinguish themselves in several aspects from the model configurations used for the CMIP6 experiments (Doescher et al., 2019):

1. The stratospheric aerosol forcing in EC-Earth3P(-HR) is handled in a simplified way that neglects the details of the vertical distribution and only takes into account the total aerosol optical depth in the stratosphere which is then evenly distributed across the stratosphere. This approach follows the treatment of stratospheric aerosols as it was used by EC-Earth2 for the CMIP5 experiments yet with the stratospheric aerosol optical depth (AOD) at 500 nm updated to the CMIP6 data set.

2. A sea surface temperature (SST) and sea-ice forcing data set specially developed for HighResMIP is used for AMIP experiments (Kennedy et al., 2017). The major differences compared to the standard SST forcing data sets for CMIP6 are the higher spatial (0.25 deg vs. 1 deg) and temporal (daily vs. monthly) resolution. For the Tier 3 HighResMIP SST forced future AMIP simulations (see section 4.1) an artificially produced data set of SST and sea ice concentration (SIC) is used that combines observed statistics and modes of variability with an extrapolated trend (https://esgfnode.llnl.gov/search/input4mips/).

3. The HighResMIP protocol requires the simulations to start from an atmosphere and land initial state from the 1950 of the ECMWF ERA-20C (Poli et al., 2016) reanalysis data. Because the soil moisture requires at least 10 years to reach equilibrium with the model atmosphere, a spin-up of 20 years under 1950 forcing has been made before starting the Tier 1 simulations.

4. In agreement with the HighResMIP protocol, the vegetation is prescribed as a present-day climatology that is constant in time.

5. The climatological present-day vegetation, based on ECMWF ERA-Interim (Dee et al., 2011), and specified as albedos and leaf area index (LAI) from the Moderate-resolution Imaging Spectroradiometer (MODIS) is used throughout all runs. In contrast, the model version for other CMIP6 experiments uses lookup table to account for changes in land-use. In addition, that version is consistent with the CMIP6 forcing data set and not based on ERA-Interim.

6. Another difference is the version of the pre-industrial aerosols background derived from the TM5 model (Van Noije et al., 2014; Bergman et al., in prep. and references therein): version 2 in PRIMAVERA, version 4 in other CMIP6 model configurations using prescribed anthropogenic aerosols. This affects mainly the sea-spray source, and in turn the tuning parameters.

## 3 Model performance and data handling

New developments in global climate models require special attention in terms of high-performance computing (HPC) due to the demand for increased model resolution, large numbers of experiments and increased complexity



of Earth System Models (ESMs). EC-Earth3P-HR (and VHR) is a demanding example where an efficient use of the resources is mandatory.

The aim of the performance activities for EC-Earth3P-HR is to adapt the configuration to be more parallel, scalable and robust, and to optimize part of the execution when this high-resolution configuration is used. The performance activities are focused on three main challenges: (1) scaling of EC-Earth3P-HR to evaluate the ideal number of processes for this configuration, (2) analyses of the main bottlenecks of EC-Earth3P-HR and (3) new optimizations for EC-Earth3P-HR.

### 3.1 Scalability

The results of the scalability analyses of the atmosphere (IFS) and ocean (NEMO) components of EC-Earth3P-HR are shown in Fig. 1, and for the fully coupled model in Fig. 2. Acosta et al. (2016) showed that, while for coupled application the load balance between components has to be taken into account in the scalability process,

the process needs to start with a scalability analysis of each individual component. However, if the optimization of one component (e.g. the reduction of the execution time of IFS) does not reduce the execution time of the coupled application, because of other slower components, a load balance analysis is required. The final choice depends on the specific problem, where either time or energy can be minimized. In section 3.2, we describe how the optimal load balance between the two components, where NEMO is the slowest component, was achieved

(Acosta et al., 2016).

### *3.2 Bottlenecks*

For the performance analysis, the individual model components (IFS, NEMO and OASIS) are benchmarked and analyzed using a methodology based on extracting traces from real executions. These traces are displayed using the Paraver software and processed to discover possible bottlenecks (Acosta et al., 2016). Eliminating these bottlenecks not only involves an adjustment of the model configuration and a balance of the number of cores devoted to each one of its components, but also modifications of the code itself and work on the parallel

programming model adopted in the different components.

The first step of a performance analysis consists in analyzing parallel programming model codes using targeted performance tools. Figure 3a illustrates an example of the performance tool's output from one single EC-Earth3P-HR model execution as provided by the Paraver tool, focusing only on its two main components: NEMO and IFS.

This figure is very useful to determine the communications within the model and identify sources of bottlenecks, especially those resulting from communication between components. It displays the communications pattern as a function of time. The vertical axis corresponds to the different processes executing the model, the top part for IFS and the lower part for NEMO. The different colors correspond to different MPI communication functions, except the light blue, which corresponds to no communication. Red, yellow and purple colors are related to MPI

communications. The green color represents the waiting time needed to synchronize the coupled model for the next time step, which means an unloaded balance in the execution. In summary, light blue areas are pure





computation and should be maximized. On the other hand, yellow, red and purple are representing overhead from parallel computation and should be minimized if possible. Additionally, green areas are preferably to be also reduced, for example increasing the number of parallel resources of the slowest component, but no optimizations are needed. From this analysis, several things can be concluded related to the overhead from parallel computation:

1) Figure 3 shows the coupling cost from a computational point of view, including one regular time step of IFS and NEMO and one time step including the coupling process. In the top part of Fig. 3a, we notice that during the first half of the first time step, the IFS component model reserves most of its processors for execution (512 processes). To simplify, it can be said that the first half of the time step has less MPI communication, with more computation-only regions, while the second half of the time step is primarily about broadcasting messages (yellow and white colour block), which corresponds to the coupling computation and to send/receive files from the atmospheric to the ocean model. These calculations impact the scalability of the code dramatically. This configuration increases the overhead when more and more processes are used and represents more than 50% of time execution when 1024 processes are used.

Additionally, this pattern of communications is repeated four times. This occurs because the different fields from IFS to NEMO are sent in three different groups, followed by an additional group of fields sent from IFS to the runoff mapper component. The communication of three different groups of fields to the same component is not taking advantage of the bandwidth of the network, thus increasing the overhead produced by MPI communications.

2) From other parts of the application, we also notice the expensive cost of the IFS output process for each time step. A master process gathers the data from all MPI sub-domains and prints the complete outputs at a regular time interval of three and six hours. During this process, the rest of processes are waiting for this step to be completed. Due to the large data volumes, this sequential process is very costly, increasing the execution time of IFS by about 30% when outputs are required.

3) The bottom part of Fig. 3a shows that the communication in NEMO is not very effective and that a large part of it is devoted to global communications, which appear in purple. Those communications belong to the horizontal diffusion routine, inside the ice model (LIM3) used in NEMO. The high frequency of communications in this routine prevented the model to scale. More information about MPI overhead of NEMO can be found in Tintó et al. (2019).

4) Due to the domain decomposition used by NEMO some of the MPI processes, which are used to run part of the ocean domain in parallel, were computing without use. This is because domain decomposition is done on a regular grid and a mask is used to discriminate between land and sea points. The mask creates subdomains of land points whose calculations are not used. This is illustrated in Fig. 4 showing a particular case in which 12 % of the depicted subdomains do not contain any sea-point.



### 3.3 New optimizations for the specific configuration

According to the profiling analysis done, different optimizations were implemented to improve the computational efficiency of the model:

1) The optimization ("opt") option of OASIS3-MCT was used. This activates an optimized global conservation transformation. Using this option, the coupling time from IFS to NEMO is reduced by 90% for EC-Earth3P-HR. This is because all-to-one/one-to-all MPI communications are replaced by global communications (gather/scatter and reduction) and the coupling calculations are done by all the IFS processes instead of only the IFS master process.

Another functionality of OASIS consists in gathering all fields sent from IFS to NEMO in a single group (Acosta et al., 2016). Coupling field gathering, an option offered by OASIS3-MCT, can be used to optimize coupling exchanges between components. The results show that gathering all the fields that use similar coupling transformations reduces the coupling overhead. This happens because OASIS3-MCT is able to communicate and interpolate all of the fields gathered at the same time.

Figure 3b shows the execution when "opt" and "gathering" options are used, with the 90% reduction in coupling time clearly visible (large green section). In the case of the first time step in the trace, the coupling time is replaced by waiting time, since NEMO is finishing its time step and both components have to exchange fields at the end of the time step.

2) For the output problem, the integration of XIOS as the I/O server for all components of EC-Earth can increase performance dramatically. XIOS is already used for the ocean component NEMO and the I/O server receiving also all the data from IFS processes and doing the output work in parallel and in an asynchronous way is the best solution to remove the sequential process when an IFS master process is required to do this work. This is being developed and will be included in the next version of EC-Earth.

3) Based on the performance analysis, the amount of MPI communications can be reduced (Tintó et al., 2019) achieving a significant improvement in the maximum model throughput. In the case of EC-Earth3P-HR, this translated into a reduction of 46% in the final execution time.

4). Using the tool ELPiN (Exclude Land Processes in NEMO) the optimal domain decomposition for NEMO has been implemented (Tintó et al., 2017), with computation of only ocean subdomains and finding the most efficient number of MPI processes. This substantially improves both the throughput and the efficiency (in case of 2048 processor cores 41% faster using 25% less resources). The increase in throughput was due to less computations and related to that less communications. In addition, ELPiN allows for the optimal use of the available resources in the domain decomposition depending on the shape and overlap of the subdomains.





### 3.4 Post-processing and data output


At the T511L91 resolution, the HighResMIP data request translates into an unprecedented data volume for EC-Earth. Because the atmosphere component IFS is originally a numerical weather prediction (NWP) model, it contains no built-in functionality for time-averaging the data stream during the simulation. The model was therefore configured to produce the requested three-dimensional fields (except radiative fluxes on model levels, which cannot be output by the IFS) on six-hourly basis and surface fields with three-hourly frequency. As a consequence, the final daily and monthly averages for instantaneous fields have been produced from sampling at these frequencies, whereas fluxes are accumulated in the IFS at every time step. Vertical interpolation to requested pressure or height levels is performed by the model itself.


For the ocean model, the post-processing is done within NEMO by the XIOS library which can launch multiple processes writing netCDF files in parallel, alleviating the I/O footprint during the model run. The XIOS configuration XML files were extended to produce as many of the ocean and sea ice variables as possible.

The combination of the large raw model output volume, the increased complexity of the requested data and the new format of the CMOR tables (Climate Model Output Rewriter, an output format in conformance with all the CMIP standards) required a major revision of the existing post-processing software. This has resulted in the development of the ece2cmor3 package. It is a python package that uses Climate Data Operators (CDO) [CDO, 2015] bindings for (i) selecting variables and vertical levels, (ii) time-averaging (or taking daily extrema), (iii) mapping the spectral and gridpoint atmospheric fields to a regular Gaussian grid and (iv) computing derived variables by some arithmetic combination of the original model fields. Finally, ece2cmor3 uses the PCMDI CMOR-library for the production of netCDF files with the appropriate format and metadata. The latter is the only supported step for the ocean output.

To speed up the atmosphere post-processing, the tool can run multiple CDO commands in parallel for various requested variables. Furthermore, we optimized the ordering of operations, performing the expensive spectral transforms on time-averaged fields wherever possible. We also point out that the entire procedure is driven by the data request, i.e. all post-processing operations are set up by parsing the CMOR tables and a single dictionary relating EC-Earth variables and CMOR variables. This should make the software easy to maintain with respect to changes in the data request and hence useful for future CMIP6 experiments.


### 4  Results

### 4.1 Outline of HighResMIP protocol


The protocol of the HighResMIP simulations consists of Tiers 1, 2 and 3 experiments, that represent simulations of different priority (1 highest, 3 lowest), and a spin-up procedure. The protocol also excludes specific tuning for the high resolution version compared to the standard resolution version. Below we give a short summary of the protocol. The experiment names in the CMIP6 data base are given in italics.




- Tier 1: Forced-atmosphere simulations 1950-2014; *highresSST-present*

The Tier 1 experiments are atmosphere only simulations forced using observed sea surface temperature for the period 1950-2014.


- Tier 2: Coupled simulations 1950-2050

The period of the coupled simulations is restricted to 100 years because of the computational burden brought about by the model resolution and the limited computer resources. The period 1950-2050 covers historical multi-decadal variability and near-term climate change. The coupled simulations consist of a spin-up, control, historical and future simulation.


**-**Spin-up simulation*; spinup-1950*

Due to the large computer resources needed, a long spin-up to (near) complete equilibrium is not possible at high resolution. Therefore, as an alternative approach an analyzed ocean state representative of the 1950s is used as the initial condition for temperature and salinity (Good et al., 2013, EN4 data set). To reduce the large initial drift a

spin-up of about 50 years is made using constant 1950s forcing. The forcing consists of greenhouse gases (GHG), including $O_3$ and aerosol loading for a 1950s (~10 year mean) climatology. Output from the initial 50 year spin-up is saved to enable analysis of multi-model drift and bias, something that was not possible in previous CMIP exercises, with the potential to better understand the processes causing drift in different models.

- Control simulation; *control-1950*

This is the HighResMIP equivalent of the pre-industrial control, but using fixed 1950s forcing. The length of the control simulation should be at least as long as the historical plus future transient simulations. The initial state is obtained from the spin-up simulation.

- Historical simulation; *hist-1950*

This is the coupled historical simulation for the period 1950-2014, using the same initial state from the spin-up as the control run.

- Future simulation; *highres-future*

This is the coupled scenario simulation 2015-2050, effectively a continuation of the hist-1950 experiment  into the future. For the future period the forcing fields are based on the CMIP6 SSP5-8.5 scenario.

- Tier 3: Forced-atmosphere 2015-2050 (2100); *highresSST-future*

The Tier 3 simulation is an extension of the Tier 1 atmosphere-only simulation to 2050, with an option to continue

to 2100. To allow comparison with the coupled integrations, the same scenario forcing as for Tier 2 (SSP5-8.5) is used.

A schematic representation of the HighResMIP simulations is given in figure 5.



### 4.2 Main results of EC-Earth3P(-HR) HighResMIP simulations

For each of the HighResMIP tiers more than one simulation was produced. An overview of the simulations is given in Table 2.


The data is stored on the JASMIN server at CEDA (http://www.ceda.ac.uk/projects/jasmin/) and available from ESGF. During the PRIMAVERA project the data was analyzed at the JASMIN server. For the *highresSST-present* and *highresSST-future* simulations the ensemble members were started from perturbed initial states. These were created by small perturbations on the three-dimensional temperature field. For the *control-1950* and *hist-1950,* the end of the spin-up was taken as the initial condition of the first member. For the two extra members the initial conditions were generated by continuing the spin-up for 5 years after perturbing the fields that are exchanged between atmosphere and ocean. The *highres-future* members are the continuation of the *hist-1950* members.

The Atlantic Meridional Overturning Circulation (AMOC) in the *control-1950* of EC-Earth3P had unrealistically low values of less than 10 Sv. It was therefore decided to change the ocean mixing parameters, which improved the AMOC. The new mixing scheme was also applied to EC-Earth3P-HR, to ensure the same set of parameters for both versions of EC-Earth3P(-HR). The simulations with the new ocean mixing are denoted with 'p2' for the coupled simulations in Table II. The atmosphere is unchanged and therefore the atmosphere simulations are denoted as 'p1'. Because of the unrealistic low AMOC in EC-Earth3P in the 'p1' simulations we focus on 'p2' for the coupled simulations.

Below we will briefly discuss the mean climate and variability of the *highresSST-present*, *control-1950* and *hist-1950* simulations. The main differences between EC-Earth3P and EC-Earth3P-HR will be highlighted. In addition the spin-up procedure for the coupled simulations, *spinup-1950*, will be outlined. A more extensive analysis of the HighResMIP simulations will be presented in forthcoming papers.

### 4.2.1 *highresSST-present*

The *highresSST-present* simulations will be compared with ERA-Interim (1979-2014) except for precipitation where GPCP V2.3 (1979-2014) (Adler et al., 2003) data will be used. Both EC-Earth and GPCP data are regridded to the ERA-Interim resolution (N128) before comparison. Seasonal means (Dec.-Feb. (DJF) and Jun.-Aug. (JJA)) will be analyzed. Ensemble mean fields will be displayed.

Due to the prescribed SST the largest surface air temperature (SAT) biases are over the continents (Fig. 6). The most negative biases are over the Sahara for DJF and Greenland in JJA while the largest positive biases are located over Antarctica in JJA and northeastern Siberia in DJF. Over most areas EC-Earth3P-HR is slightly too cold. Over most of the tropics the mean sea level pressure (MSLP) is underestimated, whereas over Antarctica and surrounding regions of the Southern Ocean it has a strong positive bias (Fig. 7). Further noteworthy is the positive bias south of Greenland during DJF. The largest precipitation errors are seen in the tropics over the warm pool regions in the Pacific and the Atlantic with too much precipitation (Fig. 8). The planetary wave structure of the



geopotential height at 500 hPa (Z500) during DJF is well represented with the exception of the region south of Greenland, which is consistent with the MSLP bias (Fig. 9).

Doubling of the atmospheric horizontal resolution has only modest impact on the large-scale structures of the main meteorological variables, as illustrated by the global MSLP, SAT, and precipitation (Fig. 10). For SAT the differences are generally less than 1 K, for MSLP 1 hPa except for the polar regions. For precipitation the difference can be larger than 1.5 mm/day in the tropics. It is possible to conclude that the increase of resolution does not have a clear positive impact on the climatology of any of those variables. For instance for precipitation it results in an increase of the wet bias over the warm pool (compare with Fig. 8).


### 4.2.2 *spinup-1950*

As discussed in the outline of the HighResMIP protocol, the spin-up was started from an initial state that is based
on observations for 1950. For the ocean this is the EN4 ocean reanalysis (Good et al., 2013) averaged over the 1950-1954 period, with 3m sea-ice thickness in the Arctic and 1m in the Antarctic. The atmosphere-land system was initialized from ERA-20C for 1950-01-01, and spun-up for 20 years to let the soil moisture reach equilibrium. For the ocean no data assimilation has been performed, which can result in imbalances between the density and velocity fields giving rise to initial shocks and waves.


During the first years of the spin-up there is a strong drift in the model climate (not shown). For the fast components of the climate system like the atmosphere and the mixed layer of the ocean the adjustment is in the order of one year, whereas the slow components such as the deep ocean require thousand years or more to reach equilibrium. For the land component this is on the order of a decade. As a consequence after a spin-up of 50 years
the atmosphere, land and upper ocean are approximately in equilibrium while the deeper ocean is still drifting. This drift also has an impact on the fast components of the climate system, which therefore still might reveal trends.

### 4.2.3 *control-1950*


After the spin-up the SAT each of the three members of EC-Earth3P-HR is in quasi-equilibrium and the global mean temperature oscillates around 13.9 °C (Fig. 11-left, black). The ocean is still warming as expressed by a negative net surface heat flux in the order of -1.5 Wm$^{-2}$ (positive is upward) (Fig. 11-right, black). This imbalance is reduced during the simulation, but without an indication that the model is getting close to its equilibrium state.


Contrary to EC-Earth3P-HR, the global annual mean SAT of EC-Earth3P displays a significant upward trend, with an indication of stabilizing after about 35 years (Fig. 11-left, red). This warming trend is caused by a large warming of the North Atlantic as revealed by Fig. 12 showing the difference between the first and last 10 years of the control-1950 run. This warming is caused by the activation of the deep convection in the Labrador Sea (not
shown), that started about 10 years after the beginning of the control simulation, which was absent in the spin-up run. Associated with that also the AMOC shows an upward trend (see Fig. 17 below). This switch to a warmer





state does not strongly affect the slow warming of the deeper ocean, which is reflected in a similar behavior of the net surface heat flux as for EC-Earth3P-HR (Fig. 11-right).

The *control-1950* experiment is also analyzed to evaluate model performance of internally-generated variability in the coupled system; the targets are: El Niño-Southern Oscillation (ENSO), the North Atlantic Oscillation (NAO), sudden stratospheric warmings (SSWs) and the Atlantic Meridional Overturning (AMOC).

*ENSO*

Figure 13 depicts the seasonal cycle of the Niño3.4 index (SST anomalies averaged over 5ºS-5ºN/170ºW-120ºW). As it was also shown for EC-Earth3.1 (Yang et al 2019), both EC-Earth3P and EC-Earth3P-HR still have a systematic underestimation of the ENSO amplitude from late-autumn to mid-winter and yield the minimum in July, 1-2 months later than in observations. Increasing model resolution reduces the bias in early-summer (May-June) but worsens it in late-summer (July-August). Overall, EC-Earth3P-HR shows lower ENSO variability than
EC-Earth3P, which following Yang et al.'s (2019) arguments suggests that the ocean-atmosphere coupling strength over the tropical Pacific is stronger in the high-resolution version of the model. On the other hand, Fig. 14 displays the spatial distribution of winter SST variability and the canonical ENSO pattern, computed as linear regression onto the Niño3.4 index. Increasing model resolution leads to a reduction in the unrealistic zonal extension of the cold tongue towards the western tropical Pacific, which was also present in EC-Earth3.1 (Yang
et al., 2019) and is a common bias in climate models (e.g. Guilyardi et al., 2009): EC-Earth3P reaches longitudes of Papua New Guinea (Fig. 14a), while EC-Earth3P-HR improves its location (Figs. 14b-c), yet overestimated as compared to observations (not shown; see Yang et al. 2019). Consistently, the improvement in the cold tongue translates into a better representation of the ENSO pattern (Fig. 14-bottom). Nonetheless, the width of the cold tongue in EC-Earth3P-HR is still too narrow in the central tropical Pacific (see also Yang et al., 2019), which
again is a common bias in climate models (e.g. Zhang and Jin, 2012).

On another matter, note that EC-Earth3P-HR (Fig. 14b) captures much better the small-scale features and meanderings along the western boundary currents, Kuroshio-Oyashio and Gulf Stream, and the sea-ice edge over the Labrador Sea than EC-Earth3P (Fig. 14a). In these three areas there is a substantial increase in SST variability
(Fig. 14c), which following Haarsma et al. (2019) is likely due to increasing ocean resolution rather than atmosphere resolution.

*NAO*

Figure 15 illustrates how EC-Earth3P(-HR) simulates the surface NAO and its hemispheric signature in the middle
troposphere. The NAO (here measured as leading EOF of the DJF SLP anomalies over 20ºN-90ºN/90ºW-40E) accounts for virtually the same fraction of SLP variance in both model versions, i.e. 42.70% in EC-Earth3P (Fig. 15c) and 42.74% in EC-Earth3P-HR (Fig. 15d), and still slightly underestimates the observed one (~50% in ERA-Interim, e.g. García-Serrano et al., 2015); the same applied to EC-Earth2.2 when compared to ERA-40 (Hazeleger et al. 2012). EC-Earth rightly captures the circumglobal pattern associated with the NAO at upper levels (e.g.
Branstator, 2002; García-Serrano and Haarsma, 2017), particularly the elongated lobe over the North Atlantic and the two centers of action over the North Pacific (Fig. 15-top). A close inspection to the barotropic structure of the NAO reveals that the meridional dipole is shifted westward in EC-Earth3P-HR (Fig. 15-right) as compared to EC-





Earth3P (Fig. 15-left), which according to Haarsma et al. (2019) could be related to increasing ocean resolution and a stronger forcing of the North Atlantic storm-track.


*SSWs*

Also the simulation of SSW occurrence is assessed (Fig. 16); the identification follows the criterion in Palmeiro et al. (2015). The decadal frequency of SSWs in EC-Earth is about 8 events per decade regardless model resolution, which is underestimated when compared to ERA-Interim (~11 events per decade) but in the range of

observational uncertainty (e.g. Palmeiro et al., 2015; Ayarzagüena et al., 2019). The same underestimation was diagnosed in EC-Earth3.1 (Palmeiro et al., 2019a). The reduced amount of SSWs is probably associated with a too-strong bias at the core of the polar vortex, still present in EC-Earth3.3 (Palmeiro et al., 2019b). It is thus concluded that increasing horizontal resolution does not affect the model bias in the strength of the polar vortex. The seasonal cycle of SSWs in reanalysis is quite robust over the satellite period, showing one maximum in

December-January and another one in February-March (Ayarzagüena et al., 2019), which was properly captured by EC-Earth3.1 in control, coupled simulations with fixed radiative forcing at year 2000 (Palmeiro et al., 2019a). Here in *control-1950*, EC-Earth does not reproduce such bimodal cycle, with EC-Earth3P-HR (blue) yielding a peak in January-February and EC-Earth3P (red) two relative maxima in January and March. Interestingly, the seasonal cycle of SSWs over the historical, pre-satellite period shows a different distribution with a prominent

maximum in mid-winter and a secondary peak in late-winter, although it is less robust among reanalysis products (Ayarzagüena et al., 2019). The impact of the (historical ozone) radiative forcing on SSW occurrence definitely deserves further research.

*AMOC*

The AMOC index was computed as the maximum stream function at 26.5N and between 900 and 1200 m depth. The annual AMOC index of EC-Earth3P-HR for the *control-1950* runs (Fig. 17-left, black) is about 15 SV, which is lower than the values form the Rapid array (Smeed et al., 2019) that are measured since 2004 (Fig. 17 stars in middle panel). It reveals interannual and decadal variability, without an evident trend. As already discussed at the beginning of section 4.2.3, the AMOC of EC-Earth3P shows an upward trend (Fig. 17-left, red) associated with

the activation of convection in the Labrador sea.

#### 4.2.4 *hist-1950*

The *hist-1950* ensemble simulations differ from the *control-1950* simulations by the historical GHG and aerosol concentrations. The global mean annual temperature in EC-Earth3P-HR displays an increase similar to the ERA-Interim data set (Fig. 18-left, black). The warming seems to be slightly larger in the model. We remind, however, the enhanced observed warming after 2014, which might result in a similar trend in the model simulations compared to observations up to present day. The cooling due to the Pinatubo eruption in 1991 is clearly visible in

all members and the ensemble mean. The amplitude and period compare well with ERA-Interim. On its part, the AMOC in EC-Earth3P-HR reveals a clear downward trend in particular from the 1990s onward (Fig. 17-middle, black). This is consistent with a slowdown of the Atlantic overturning due to global warming in CMIP5 models (Cheng et al., 2013).



Similarly to control-1950, the hist-1950 simulations with EC-Earth3P show upward trends in SAT (Fig. 18-left, red) and AMOC (Fig. 17-middle, red) that are smaller (SAT) or absent (AMOC) in EC-Earth3P-HR. The HighResMIP protocol (Haarsma et al., 2016) of having a control and a historical simulation starting from the same initial conditions was designed to minimize the consequences of such trends. Under the assumption that the model trend is similar for both simulations, it can be eliminated by subtracting the control from the historical simulation.

Indeed the global annual mean SAT and the AMOC of hist-1950 – control-1950 display a very similar behavior in EC-Earth3P and EC-Earth3P-HR (Fig. 18-right and Fig. 17-right) with an upward trend for SAT and a downward trend for the AMOC. For SAT the upward trend compares well with ERA-interim.

*Weather regimes*

Another way to test the representation of the mid-latitude atmospheric flow, with a focus on the low frequency variability (5 – 30 days), is to assess how well the models reproduce the winter (DJF) Euro-Atlantic weather regimes (Corti et al., 1999; Dawson et al., 2012).

The analysis has been applied here to the EC-Earth3P and EC-Earth3P-HR *hist-1950* simulations. Following recent works (Dawson, 2015; Strommen, 2019), we computed the regimes via k-means clustering of daily geopotential height anomalies at 500 hPa over 80W-40E/30N-85N. As a reference, we considered the ECMWF reanalysis data from ERA40 (1957-1978) and ERA-Interim (1979-2014). The clustering is performed in the space spanned by the first 4 Principal Components obtained from the reference dataset. More details on the technique

used and on the metrics discussed here can be found in Fabiano et al. (to be submitted) and references therein. Each row in Fig. 19 shows the resulting mean patterns of the four standard regimes - NAO+, Scandinavian Blocking, Atlantic Ridge and NAO- - for ERA (top), EC-Earth3P (middle) and EC-Earth3P-HR (bottom). The regimes are quite well represented in both configurations. However, the matching is better in the standard resolution version both in terms of RMS and pattern correlation averaged over all regimes (see Table 3). Only the

Scandinavian (Sc) blocking pattern is improved in the HR, whereas the other patterns are degraded. The most significant degradation is seen for the NAO- pattern, which is shifted westward in the HR simulation. The result for EC-Earth3P(-HR) goes in the opposite direction of what has been observed in Fabiano et al. (to be submitted), where most models showed a tendency for improving the regime patterns with increased resolution. Concerning the regime frequencies, both model versions show a tendency to produce less NAO+ cases than the observations

and more Atlantic Ridge cases (Fig. 19).

Another quantity of interest is the persistence of the regimes, since models usually are not able to reach the observed persistence of the NAO+/- states (Fabiano et al., to be submitted). As stated in Table 3, this is also observed for the EC-Earth3P *hist-1950* simulations and the effect of the HR is to increase the persistence of

NAO+, but decrease that of NAO-.

Even if the HR is degrading the regime patterns, it produces a small but positive effect on the geometrical structure of the regimes. This is shown by the last two quantities in Table III: the optimal ratio and the sharpness. The optimal ratio is the ratio between the mean inter-cluster squared distance and the mean intra-cluster variance: the





larger the optimal ratio, the more clustered are the data. The sharpness is an indicator of the statistical significance
       of the regime structure in the dataset in comparison with a randomly sampled multinormal distribution (Straus et
       al., 2007). The closer the value is to 100, the more significant is the multimodality of the distribution. The
       sharpness tends to saturate at 100 for very long simulations, so the values reported in Table 3 are obtained from a
       bootstrap on 30 years randomly chosen. Both the optimal ratio and the sharpness are too low in the EC-Earth3P
simulations, as is usually seen for all models. A significant increase with EC-Earth3P-HR is seen for the optimal
       ratio, and a smaller (non-significant) one is seen for the sharpness.

       The increased resolution simulations have a stronger regime structure and are closer to the observations in this
       sense. However, the regime patterns are degraded in the HR version and this affects the resulting atmospheric
flow. A similar result was obtained by Strommen et al. (2019), for a different version of EC-Earth and two other
       climate models.

## 5 Discussion and conclusions


       As contribution of the EC-Earth consortium to HighResMIP, a new version of EC-Earth has been developed with
       two horizontal resolutions: the standard resolution EC-Earth3P (T255, ORCA1) and the high-resolution EC-
       Earth3P-HR (T511, ORCA0.25). Simulations following the HighResMIP protocol (Haarsma et al., 2016) for all
       three tiers have been made using both resolutions, with an ensemble size of three members. Only the spin-up
consists of one member.

       Performing 100-yr simulations for the high-resolution version (EC-Earth3P-HR) required specific developments
       for the hard and soft ware to ensure efficient production, post-processing and storage of the data. In addition, the
       model must be able to run on different platforms with similar performance. Large efforts have been dedicated to
scalability, reducing bottlenecks during performance, computational optimization and efficient post-processing
       and data output.

       Enhancing resolution does not noticeably affect most model biases and there are even locations and variables
       where increasing the resolution has a deteriorating effect such as an increase of the wet bias over the warm pool
seen in the *highresSST-present* simulations or the representation of Euro-Atlantic weather regimes found in the
       hist-1950 experiments. Also the variability reveals examples of improvement such as the zonal extension of the
       ENSO pattern or the representation of meandering along the western boundary currents, as revealed in the control-
       1950 simulations. The lack of re-tuning the high-resolution version of the model compared to the standard-
       resolution version, in accordance with the HighResMIP protocol, might be responsible for this.


       The short spin-up as prescribed by the HighResMIP protocol prevented the simulations to reach an equilibrium
       state. This happened in particular for the *control-1950* and *hist-1950* simulations of EC-Earth3P where a transition
       to a warmer state occurred due to enhanced convection in the Labrador Sea, with an accompanying increase of
       the MOC. Because this transition occurred almost concurrently in the *control-1950* and *hist-1950* simulations the



greenhouse-forced warming from 1950 onward in EC-Earth3P can be inferred by subtracting both simulations. The resulting warming pattern compares well with the observed one and is similar to the warming pattern simulated by EC-Earth3P-HR. Due to the transition, the *control-1950* does not provide a near-equilibrium state. It was therefore decided to extend the *control-1950* run for another 100 year to allow process studies, that will be documented elsewhere.


Analysis of the kinetic energy spectrum indicates that the sub-synoptic scales are better resolved at higher resolution (Klaver et al., 2019) in EC-Earth. Despite the lack of a clear improvement with respect to biases and synoptic scale variability for the high resolution version of EC-Earth, the better representation of sub-synoptic scales results in better representation of phenomena and processes on these scales such as tropical cyclones

(Roberts et al., 2019) and ocean-atmosphere interaction along western boundary currents (Tsartsali et al. in preparation).

**Code/data availability,**

Model codes developed at ECMWF, including the IFS and FVM, are intellectual property of ECMWF and its m

ember states. Permission to access the EC-Earth source code can be requested via the EC-Earth website http://www.ec-earth.org (The EC-Earth Consortium, 2019) and may be granted if a corresponding software licence agreement is signed with ECMWF. The repo tags for the versions of IFS and EC-EARTH that are used in this work are 3.0p (see section 4.2, "p1" version) and 3.1p ("p2" version), and are available through r7481 and r7482 on ECSF respectively. The model code evaluated in this manuscript has been provided for

anonymous review by the topical editor and anonymous reviewers.

The doi of the data used in the analyses and available from ESGF are:
EC-Earth3P doi:10.22033/ESGF/CMIP6.2322
EC-Earth3P-HR doi:10.22033/ESGF/CMIP6.2323


**Author contributions,** RH, MA, PAM, LPC, MC, SC, PD, FB, JGC, TK, VM, TvN, FMMP, PLS, MvW, KW contributed to the text and the analyses. All authors contributed to the design of the experiment, model development, simulations and post-processing of the data.

**Competing interests,** the authors declare that they have no competing interests.

**Financial support,**

The authors acknowledge funding received from the European Commission under Grant Agreement 641727 of the Horizon 2020 research programme.


The research leading to these results has received funding from the EU H2020 Framework Programme under grant agreement n° 748750.





This project has received funding from the European Union's Horizon 2020 research and innovation programme under the Marie Skłodowska-Curie grant agreement INADEC No 800154.

This project has received funding from the European Union's Horizon 2020 research and innovation programme under the Marie Skłodowska-Curie COFUND grant agreement No. 754433.

The EC-EARTH simulations from SMHI were performed on resources provided by the Swedish National Infrastructure for Computing (SNIC). The EC-EARTH simulations from BSC were performed on resources provided by ECMWF and the Partnership for Advanced Computing in Europe (PRACE; MareNostrum, Spain).

FMP and JG-S were partially supported by the Spanish GRAVITOCAST project (ERC2018-092835) and 'Ramón y Cajal' programme (RYC-2016-21181), respectively; whereas MR was supported by a 'Beca de col·laboració amb la Universitat de Barcelona' (2019.4.FFIS.1).





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





| Forcing | Dataset | Version |
|---|---|---|
| Solar | http://solarisheppa.geomar.de/solarisheppa/cmip6 | 3.1 |
| Well-mixed GHG concentrations | CMIP6_histo_mole_fraction_of_XXX_in_air_input4MIPs_gr1-GMNHSH.nc from input4mips with XXX being carbon_dioxide, cfc11eq, cfc12, methane or nitrous_oxide | 1.2.0 |
| Tropospheric aerosols | Anthropogenic part: MACv2.0-SP_v1.nc Pre-industrial part: based on TM5 | 2.0 |
| Stratospheric aerosols | Simplified approach. CMIP6 stratospheric AOD at 550 nm, vertically integrated | 2.1.0 |
| Ozone | vmro3_input4MIPs_ozone_CMIP6_UReading-CCMI from input4mips | 1.0 |
| Vegetation | Present day climatology. Vegetation type and cover from ERA-Interim. Albedo and LAI derived from MODIS. Same procedure as used for ERA-20C | |
| AMIP SST + SIC | HadISST2 from input4mips | 2.2.0.0. |


**Table 1** CMIP6 forcing details.






|  | *highresSST-present* | *highresSST-future* | *control-1950* | *hist-1950* | *highres-future* |
|---|---|---|---|---|---|
| **EC-Earth3P-HR** | 3 members:<br>r1i1p1f1<br>r2i1p1f1<br>r3i1p1f1 | 3 members:<br>r1i1p1f1<br>r2i1p1f1<br>r3i1p1f1 | 4 members:<br>r1i1p1f1<br>r1i1p2f1<br>r2i1p2f1<br>r3i1p2f1 | 4 members:<br>r1i1p1f1<br>r1i1p2f1<br>r2i1p2f1<br>r3i1p2f1 | 4 members:<br>r1i1p1f1<br>r1i1p2f1<br>r2i1p2f1<br>r3i1p2f1 |
| **EC-Earth3P** | 3 members:<br>r1i1p1f1<br>r2i1p1f1<br>r3i1p1f1 | 3 members:<br>r1i1p1f1<br>r2i1p1f1<br>r3i1p1f1 | 4 members:<br>r1i1p1f1<br>r1i1p2f1<br>r2i1p2f1<br>r3i1p2f1 | 4 members:<br>r1i1p1f1<br>r1i1p2f1<br>r2i1p2f1<br>r3i1p2f1 | 4 members:<br>r1i1p1f1<br>r1i1p2f1<br>r2i1p2f1<br>r3i1p2f1 |

**Table 2** Overview of the HighResMIP simulations of EC-Earth3P-HR and EC-Earth3P.


|  | ERA | EC-Earth-3P | EC-Earth3P-HR |
|---|---|---|---|
| RMS (mean) | / | 18 +/- 8 m | 22 +/- 8 m |
| Patt. corr. (mean) | / | 0.90 +/- 0.08 | 0.86 +/- 0.11 |
| Av. persistence (NAO +) | 6.1 days | 5.4 +/- 0.2 days | 5.7 +/- 0.5 days |
| Av. persistence (NAO –) | 7.0 days | 6.0 +/- 0.2 days | 5.5 +/- 0.3 days |
| Optimal ratio | 0.841 | 0.759 +/- 0.010 | 0.771 +/- 0.007 |
| Significance (30 yr) | 95.6 | 80.2 +/- 6.0 | 82.3 +/- 0.4 |

**Table 3** Some metrics to assess the overall performance in hist-1950 of the EC-Earth3P and EC-Earth3P-HR simulations in terms of weather regimes. The table shows: the average RMS deviation from the observed patterns and the relative average pattern correlation among all regimes; the average persistence of the two NAO states in days; the optimal ratio and the sharpness. The errors refer to the spread between members (standard deviation).




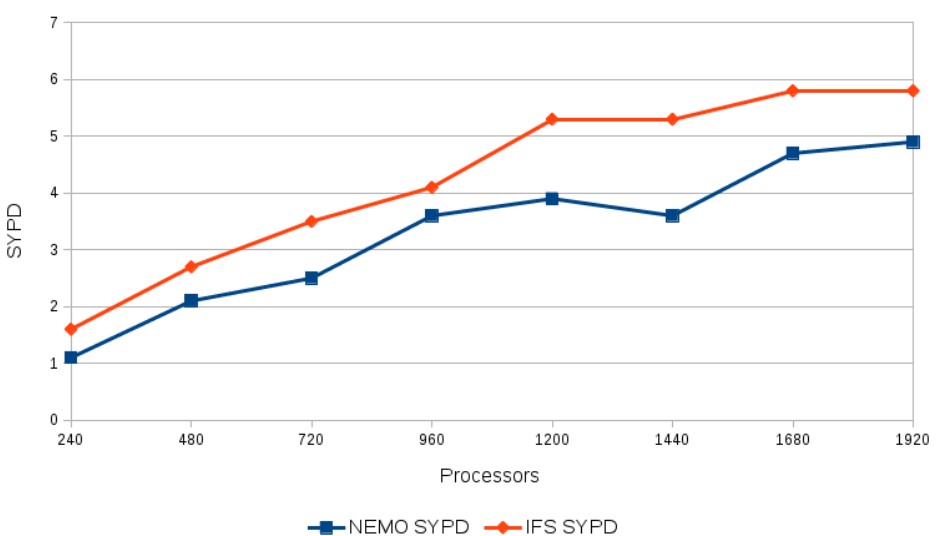

**Figure 1** NEMO (red) and IFS (blue) scalability in EC-Earth3P-HR. The throughput is expressed in simulated years per day (SYPD) of wall clock time. The tests have been performed on the MareNostrum4 computer at the

Barcelona Computing Centre with full output and samples of five one-month runs for each processor combination, the average of which is shown in the figure. The horizontal axis corresponds to the number of cores used.

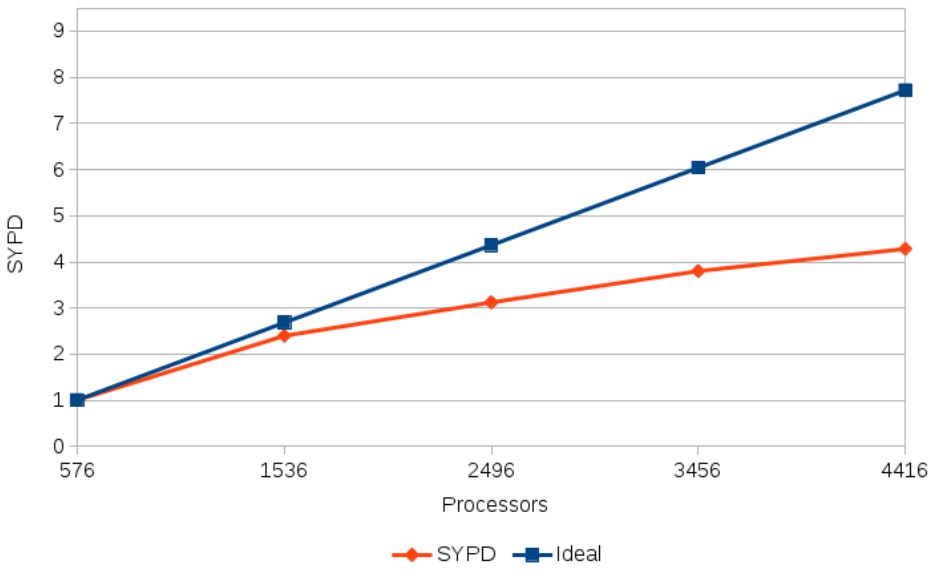


**Figure 2** As Fig. 1 but now for the scalability of the fully coupled EC-Earth3P-HR. The blue diagonal indicates perfect scalability.



a)

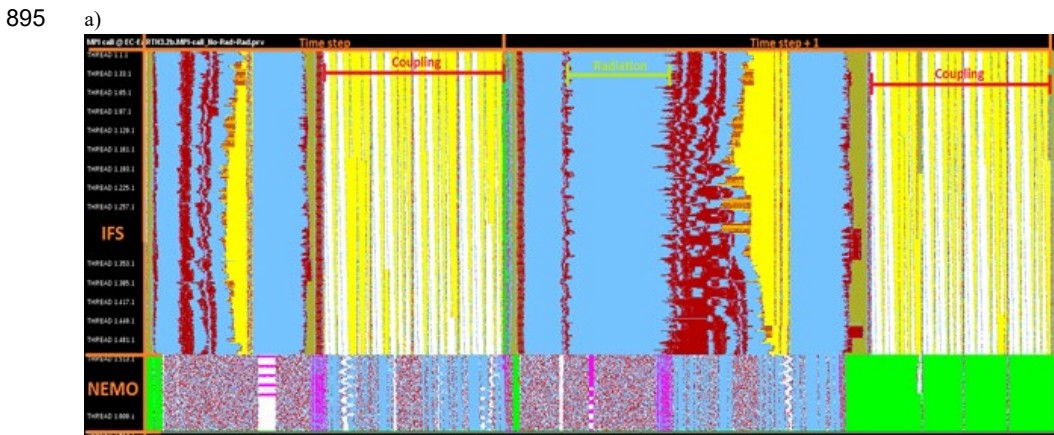

b)

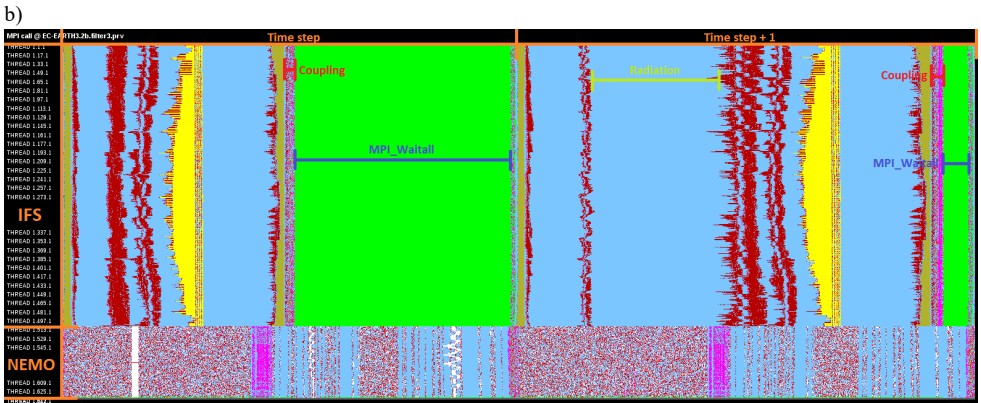


**Figure 3** (a) Paraver view of the NEMO and IFS components in an EC-Earth3P-HR model execution for two time steps including the coupling process. The horizontal lines give the behaviour of the different processes (1 to 512 for IFS and 513 to 536 for NEMO) as a function of time. Each colour corresponds to a different MPI communication function. See text for explanation. (b) as (a), but now when optimization options "opt" and 905   "gathering" for coupling are activated.







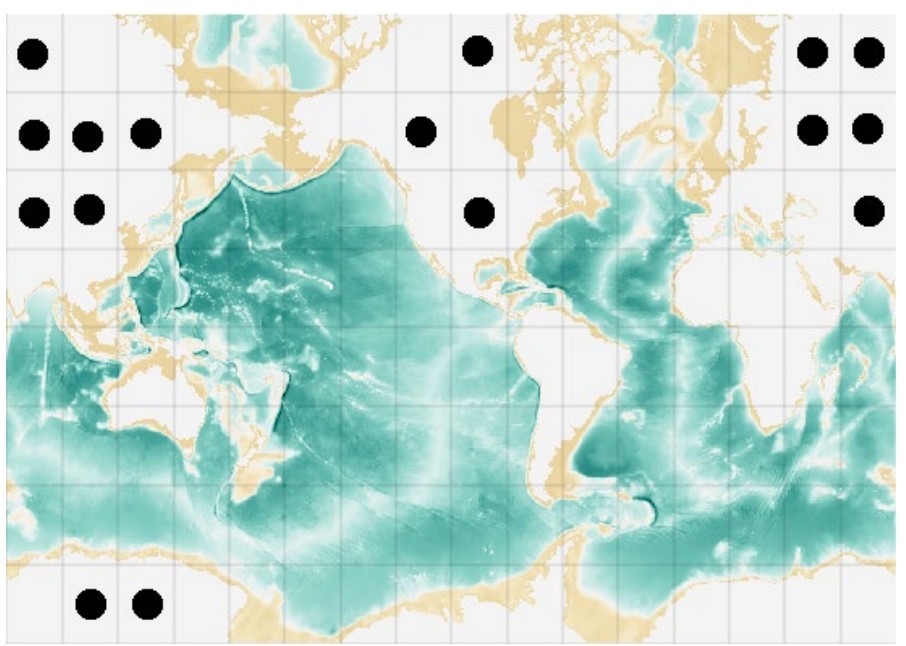

**Figure 4** Domain decomposition of a tripolar grid of the ORCA family with a resolution of a of degree into 128 subdomains (16 x 8). Subdomains marked with a black dot do not contain any ocean grid point.


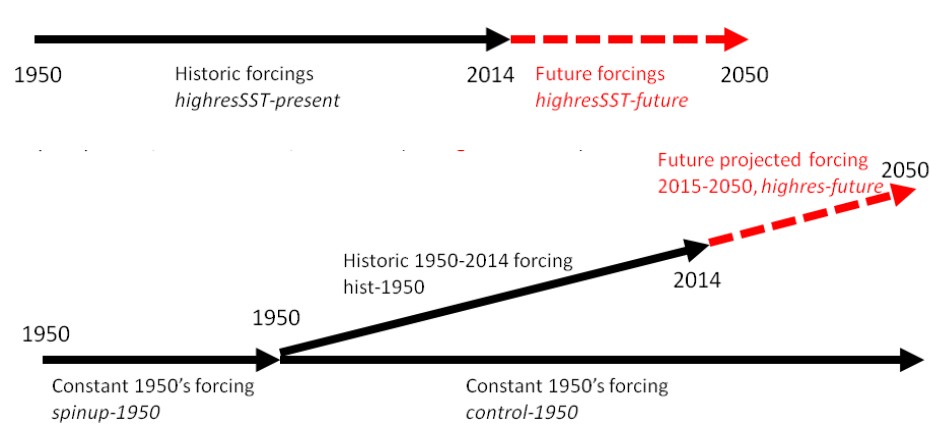

**Figure 5** Schematic representation of the HighResMIP simulations.




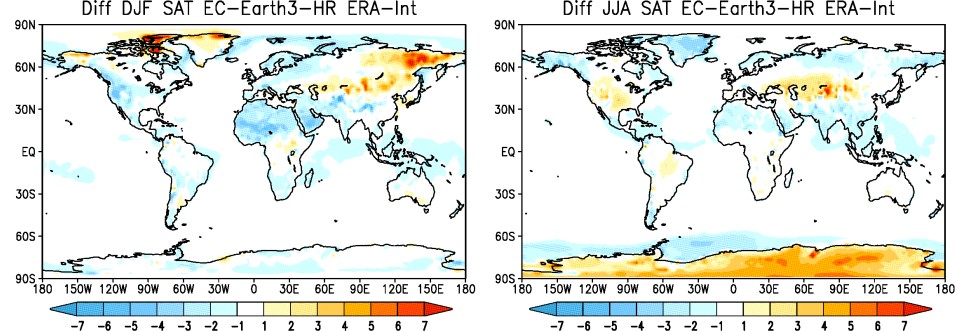

**Figure 6** SAT: Bias [ºC] of EC-Earth3P-HR with respect to ERA-Interim for the period 1979-2014. (a) DJF, (b) JJA.


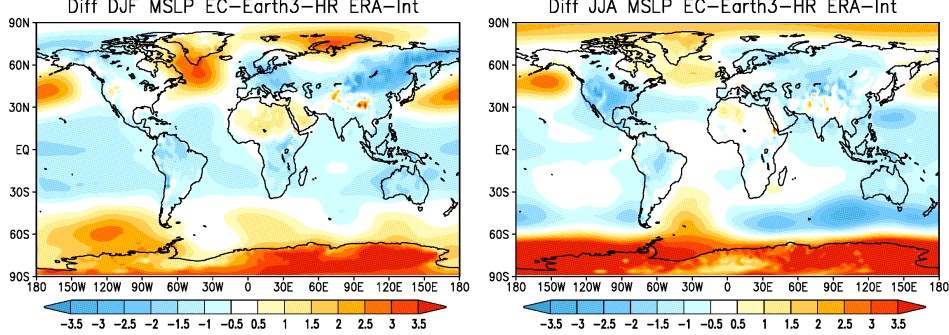

**Figure 7** MSLP: Bias [hPa] EC-Earth3P-HR with respect to ERA-Interim for the period 1979-2014. (a) DJF, (b) JJA.

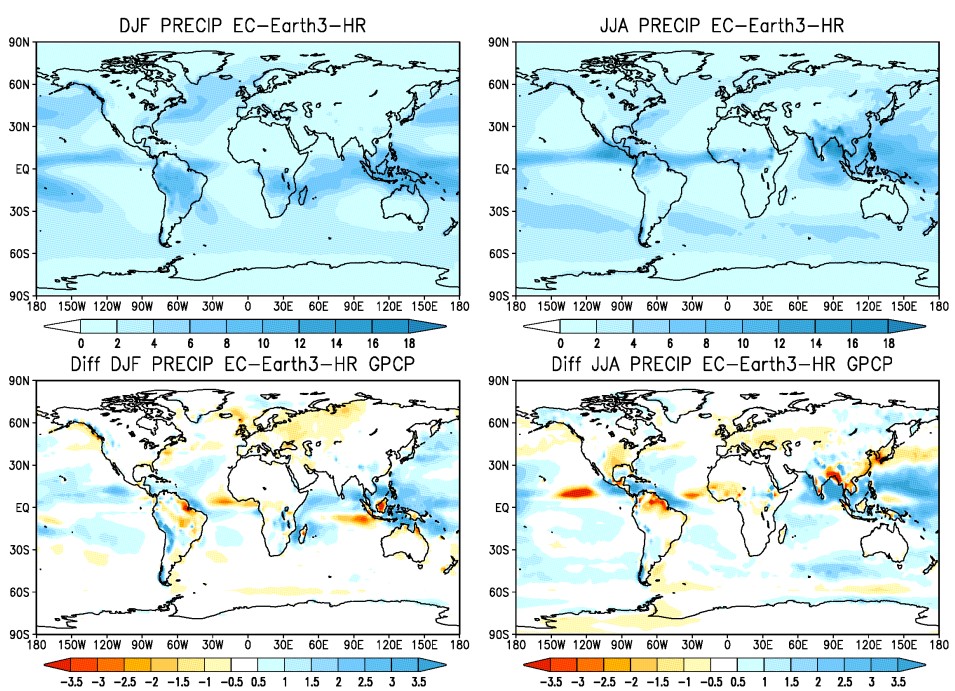

**Figure 8** Precipitation and bias EC-Earth3P-HR with respect to GPCP [mm/day] for the period 1979-2014. (a),
(c) DJF, (b), (d) JJA.

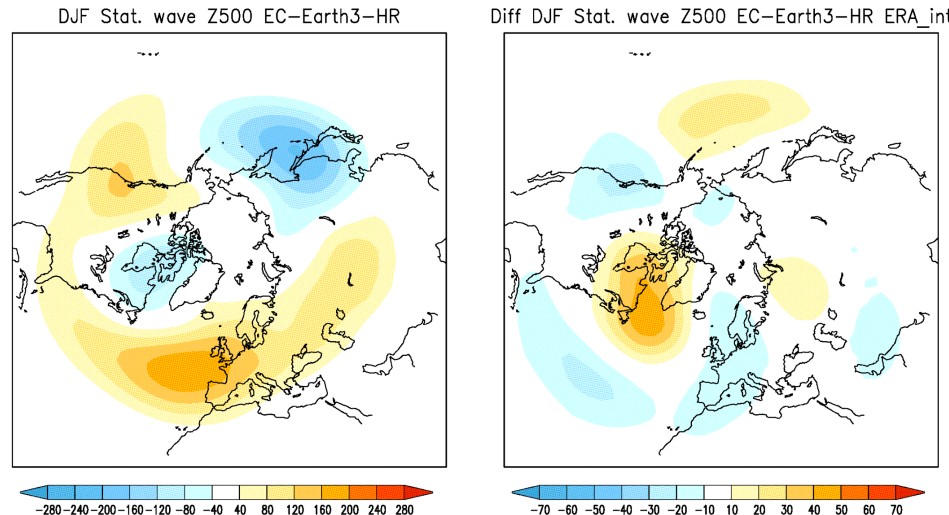

**Figure 9** (a) Stationary eddy component (departure from zonal mean) of EC-Earth3P-HR of the 500-hPa
geopotential height (m) in boreal winter; (b) the difference with ERA-Interim. Note the difference in color scale
between the two panels.






965 .


**Figure 10** Differences between EC-Earth3P-HR and EC-Eart3P for SAT [°C] (upper row), MSLP [hPa] (middle row) and precipitation [mm day-1] (bottom row), for DJF (left panels) and JJA (right panels)



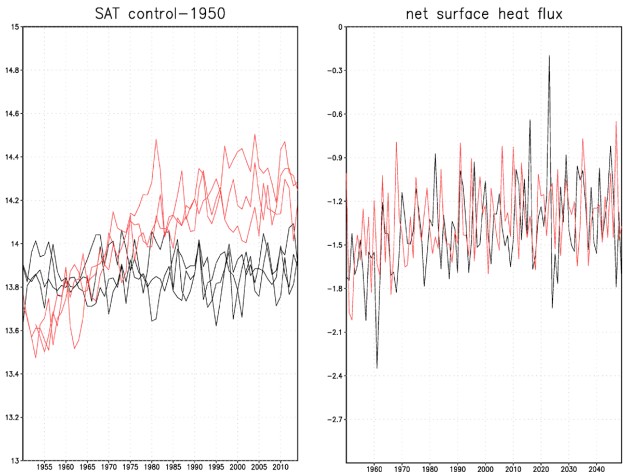

**Figure 11** Left: Global mean averaged annual SAT [°C] in control-1950 for the three members of EC-Earth3P (red) and EC-Earth3P-HR (black). Right: Global mean averaged net surface heat flux [Wm-2] in control-1950 of EC-Earth3P (red) and EC-Earth3P-HR (black), displayed only for one member (r1i1p2f1) of each model for clarity; other members display similar behavior.

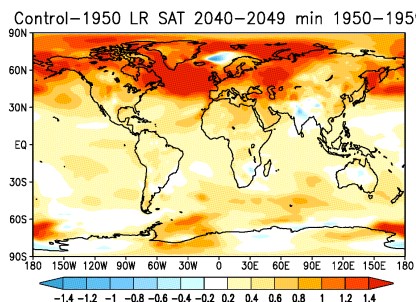

**Figure 12** Ensemble mean difference in SAT [°C] averaged over the first and last 10 years of the control-1950 simulations of EC-Earth3P.





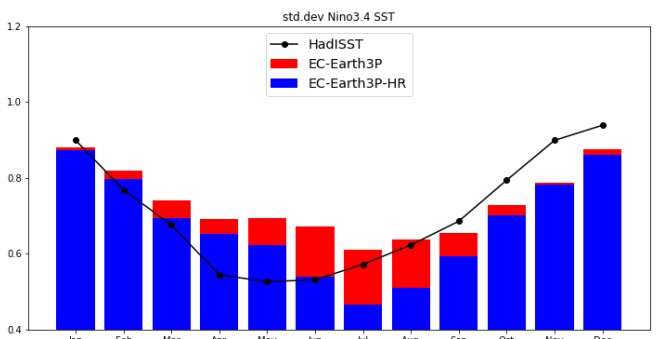

**Figure 13** Monthly standard deviation of the Niño3.4 SST index: EC-Earth3P (red) EC-Earth3P-HR (blue) from control-1950, and detrended HadISST over 1900-2010 (black).



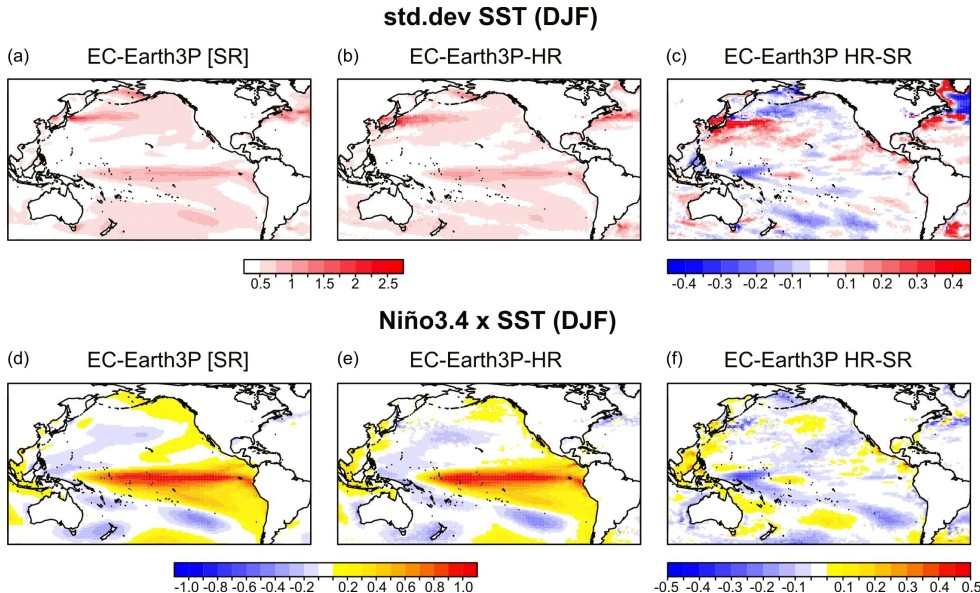

**Figure 14** Top: Boreal winter SST standard deviation from control-1950 in EC-Earth3P (a), EC-Earth3P-HR (b) and their difference (c). Bottom: Regression of SST anomalies onto the Niño3.4 index from control-1950 in EC-Earth3P (d), EC-Earth3P-HR (e), and their difference (f).


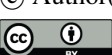

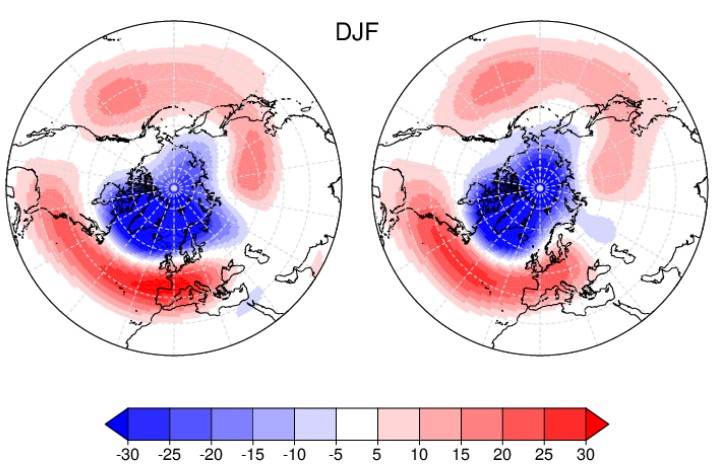

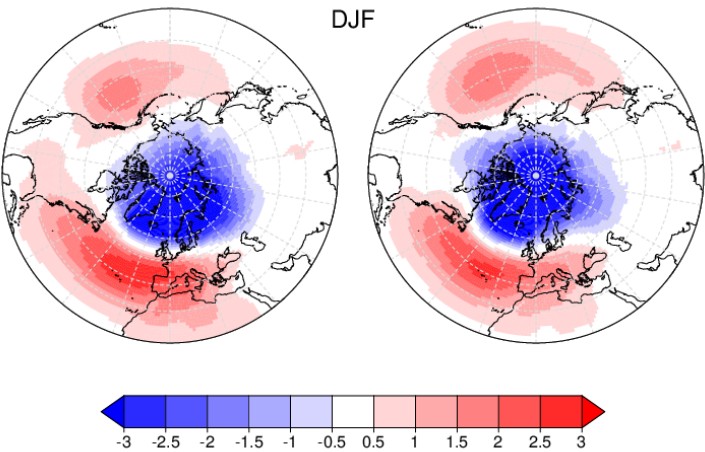


**Figure 15** Bottom: Leading EOF of winter SLP anomalies over the North Atlantic-European region 20ºN-90ºN/90ºW-40E from control-1950 in EC-Earth3P (c) and EC-Earth3P-HR (d); the corresponding fraction of explained variance is indicated in the title. Top: Regression of 500hPa geopotential height anomalies from control-1950 in EC-Earth3P (a) and EC-Earth3P-HR (b) onto the corresponding leading principal component, i.e. NAO index.



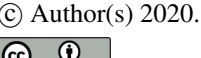

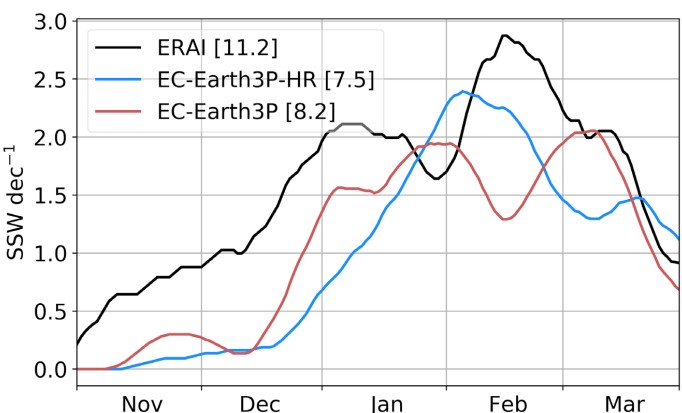

**Figure 16** Seasonal distribution of SSWs per decade in a [-10, 10]-day window around the SSW date for ERA-Interim (black), EC-Earth3P (red) and EC-Earth3P-HR (blue) from control-1950. Time-series are smoothed with a 6-day running-mean. The total decadal frequency of SSWs is indicated in brackets.

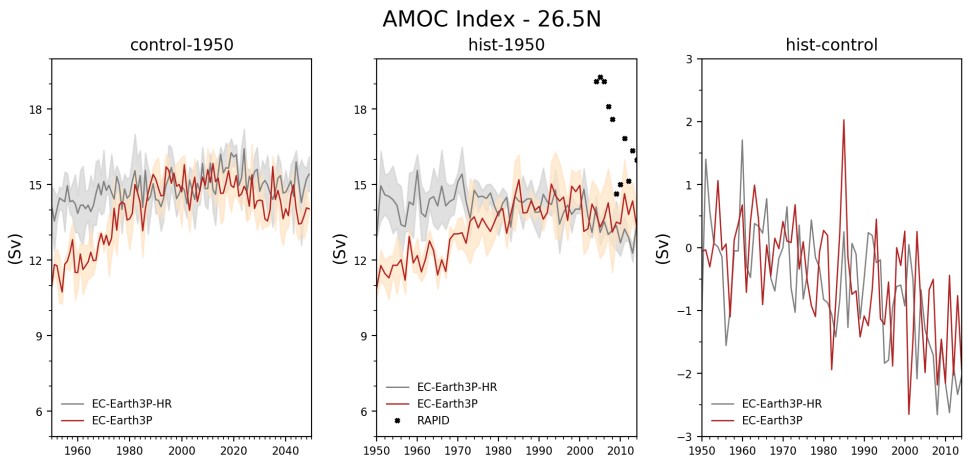

**Figure 17** Time series of the annual AMOC index for the control-1950 (left) and hist-1950 (middle) runs. Solid lines display the ensemble mean for the EC-Earth3P (red) and EC-EarthP-HR (black). Shaded areas represent the dispersion due to the ensemble members. Black stars in the middle panel displays values of RAPID data. Right: Mean ensemble difference between hist-1950 and control-1950 for Earth3P (red) and EC-Earth3P-HR (black).




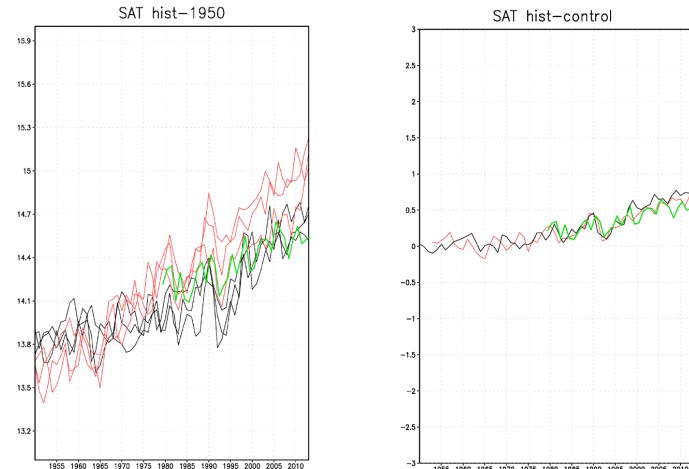


**Figure 18** Global mean averaged annual SAT [°C] in hist-1950 (left) for the three members of EC-Earth3P (red) and EC-Earth3P-HR (black). Right: Mean ensemble difference between hist-1950 and control-1950 for EC-Earth3P (red) and EC-Earth3P-HR (black). ERA-Interim is indicated by the green curves. For the right plot it is scaled so that the starting point fits with the EC-Earth curves.




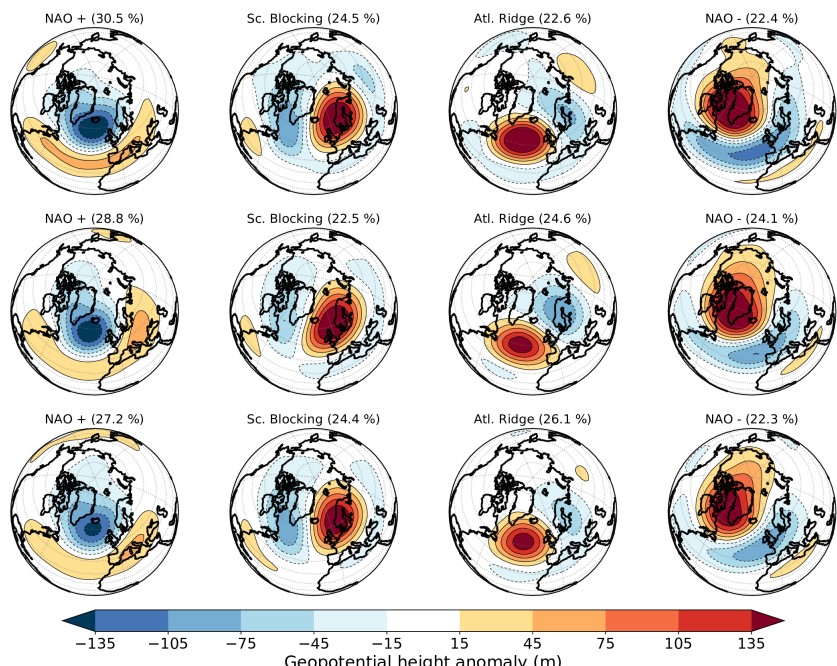

**Figure 19** Observed cluster patterns for ERA (top), simulated cluster patterns in hist-1950 for EC-Earth3P (middle) and EC-Earth3P-HR (bottom). The frequency of occurrence of each regime is shown above each subplot.
