# Peer review of "HighResMIP versions of EC-Earth: EC-Earth3P and EC-Earth3P-HR. Description, model computational performance and basic validation"

_Geoscientific Model Development, 2019_

## Referee Comment (RC1) · Anonymous Referee #1 · 14 Apr 2020

This paper describes the EC-Earth3P and EC-Earth3P-HR models developed for the HighResMIP with a lot of details, including optimization (necessary for high-res modeling) technical aspects of scalability, performance, data-storage, and post-processing and documentation of model performance regarding the mean climatology as well as variabilities. The manuscript is generally well-organized and clearly written.

General comments:

My concern is that I feel it belongs to the "Model description paper" category instead of the "Model evaluation paper". The model results seem not the primary focus and are mainly presented in a documentation manner without more in-depth analysis and

scientific insights. As stated in the middle of the text, more extensive analysis will be shown in a future paper. I suggest the authors revise it to better fit the criteria of the model description type and leave more results and analysis in the other paper.

Specific comments:

Title: The "model performance" can mean either computational performance or the quality of simulation results. Putting in the middle of "description" and "data handling", it sounds more of the former, so perhaps change it to "computational performance". Also change "validation" to "initial validation" to coordinate with the second paper?

L87: seems a good place to add resolution info since that info is given for Earth3P-VHR on L89.

L100: temporal resolutions, time steps?

L175-177: It can be a bit misleading to imply the optimization of components and load balance are purely sequential. In practice, they can be parallel, for example in the incidents that component optimization is only possible with a load rebalance.

Figure 2: change the label "SYPD" to "coupled EC-Earth3P-HR"?

L216: Where on Figure 3 can we see the 4 times of communication pattern?

L225: I am not sure what parts on Figure 3 this paragraph refers to. Please clarify.

Figure 3: This figure is too noisy. Perhaps, the authors can replot it to better support the points they want to make with improved labels, organization, and clarity.

Figure 5: change "hist-1950" to italic

L383: . . . in Table 2

Figure 6: Add the global means and RMS errors (which give some overall ideas about the model performance) and discuss these numbers in the text. Change the title of figures to, for example, ". . .EC-Earth3-HR minus ERA-Int" to be clearer. Add labels (a)

and (b). Also make these changes on other figures where applicable.

L408: change to "...Greenland (Fig. 9), which is ... MSLP bias (Fig. 7a)."

L413: I would wonder whether enhancing horizontal resolution has a negative impact on performance. The global mean biases and RMS errors (suggested above) are helpful to provide some quantitative measure.

L430: Perhaps can add some figures to support this point.

L444: Any explanations why this activation of deep convection at the Labrador Sea occurs in the low-res version, but not in the high-res version?

Figure 11: I suggest using different colors for different simulations, but similar ones for the same resolution – redish for low-res; blackish for high-res. I also suggest the authors add a panel of net radiation fluxes at the top of the atmosphere to show the energy balance of the whole Earth system.

Figure 13: Isn't it clearer to compare if the model results are shown in the same manner (lines instead of bars) as the observation? I find it is difficult to follow the seasonal cycle of EC-Earth3P – The base changes every month. Please revise it.

L534: trend -> drift? The trend on this line has a different meaning than the ones towards the end of the paragraph, so should use different words to distinguish.

L535: change to "...hist-1950 minus control-1950..."

Figure 18: I understand the authors scale the right panel to fit the starting point of the curves. But it looks a bit weird to leave large white margins on it. Please revise.

---

## Referee Comment (RC2) · Anonymous Referee #2 · 20 Apr 2020

In the present manuscript, the authors summarized basic model performance/drifts of EC-Earth3P-HR in comparison with lower resolution version, EC-Earth3P, together with optimization procedure of model code, data handling, how to post-process. The manuscript is well-organized, and basically I consider that the present manuscript will be worth publishing in GMD. In general, however, physical explanations on causes of model biases, drifts, and differences between EC-Earth3P-HR and EC-Earth3P are quite limited throughout the manuscript. After minor revisions in order to answer the suggestions and comments listed below, the manuscript will be more suitable for publication.

[Figure]

Comments

L. 373-374: How did you generate atmospheric temperature perturbations? Gaussian random noise with a certain amplitude? Please specify the method.

L. 380: How did you change the oceanic mixing parameters? Please describe more details.

L. 399-408: There are almost no explanations on cause of model biases described here. Please give possible reasons or speculations for the biases which may be arisen from, for example, deficiencies in parameterizations for cloud microphysics, (deep, shallow, strat) cumulus, insufficient horizontal resolution, albedo parameterization of snow, sea-ice, etc.

L. 411: Why the MSLP over Antarctica is higher (worse) in EC-Earth3P-HR than EC-Earth3P? Please give possible reasons or speculations for the biases. In addition, if the biases in stationary eddies (Fig. 9) MSLP (Figs. 7 and 10) are evaluated, you may want to show wintertime storm track activity defined as subweekly eddy meridional temperature flux at the 850 hPa for EC-Earth3P-HR than EC-Earth3P, which may be useful for interpreting differences of model biases between two models.

Figure 6, 7, and 8: In order to evaluate model errors quantitatively, please calculate root-mean-squared errors (RMSE) for EC-Earth3P-HR with respect to observations/reanalysis and add the RMSE to somewhere in the corresponding figures, for example, just right of the figure title as "Diff DJF SAT EC-Earth3-HR ERA-Int (0.8 K)".

Figure 10: Panels showing difference between EC-Earth3P-HR and EC-Earth3P may be replaced by the errors between EC-Earth3P and observations/reanalysis as in Figs. 6-8. And, RMSEs for EC-Earth3P may be given.

Figure 12: Top label "2040-2049 minus 1950-1959" may be wrong.

L. 444: Why does not the activation of deep convection in EC-Earth3P-HR occur and why do global-mean SAT and AMOC transport keep stable in EC-Earth3P-HR?

Figure 14: For comparison, Figs. 14c and 14f may be replaced by observations. Also Z500 anomalies regressed onto NINO3.4 index, which can be superimposed onto Fig. 14de by contours, are useful for evaluating atmospheric teleconnection pattern.

L. 473-476: SST variability is closely related to the frontal structure seen in the climatic-mean SST. So, you may want to add DJF climatic-mean SST to Figs. 14a-c by contours.

Figure 15: Again, please add the corresponding panels for observations/reanalysis.

L. 512: Please capitalize "Rapid".

L. 525: ERA-Interim is just reanalysis data, not observations. You may want to redraw the green lines in Fig. 18 based on observations, for example, HadCRUTv4.4 with keeping consistency of undefined grid point between observations and models. And then, please rewrite Section 4.2.4.

---

## Author Comment (AC1) · 11 Jun 2020

Response to Referee #1

This paper describes the EC-Earth3P and EC-Earth3P-HR models developed for the HighResMIP with a lot of details, including optimization (necessary for high-res modeling) technical aspects of scalability, performance, data-storage, and post-processing and documentation of model performance regarding the mean climatology as well as variabilities. The manuscript is generally well-organized and clearly written.
We thank the reviewer about the positive comment on organization and how it is written.

My concern is that I feel it belongs to the "Model description paper" category instead of the "Model evaluation paper". The model results seem not the primary focus and are mainly presented in a documentation manner without more in-depth analysis and scientific insights. As stated in the middle of the text, more extensive analysis will be shown in a future paper. I suggest the authors revise it to better fit the criteria of the model description type and leave more results and analysis in the other paper.
We understand the concern of the reviewer about the category of the paper. We have, however, chosen for the model evaluation category because apart from the description of the model. We provide analysis of the climatology, biases, trends, and the dominant modes of variability such as NAO, ENSO and the AMOC. Due to the space limitation of an article, these analyses are not performed in full depth and might be further analysed in forth coming papers. However, covering this wide range of aspects and phenomena we consider that the editorial board would agree on accepting the manuscript as an evaluation of the model, which indeed could serve as a starting point for further in-depth research.

Specific comments:
Title: The "model performance" can mean either computational performance or the quality of simulation results. Putting in the middle of "description" and "data handling", it sounds more of the former, so perhaps change it to "computational performance". Also change "validation" to "initial validation" to coordinate with the second paper?
Good suggestions. We have changed the title to: **HighResMIP versions of EC-Earth: EC-Earth3P and EC-Earth3P-HR. Description, model computational performance and basic validation**

L87: seems a good place to add resolution info since that info is given for Earth3P-VHR on L89.
We have added the resolution info for EC-Earth3P and EC-Earth3P-HR.

L100: temporal resolutions, time steps?
IFS and NEMO have the same time steps: 45 min in the standard configuration and 15 min at HiRes. The coupling between the model is 45 min in both resolutions. We have included that in the manuscript at L125.

L175-177: It can be a bit misleading to imply the optimization of components and load balance are purely sequential. In practice, they can be parallel, for example in the incidents that component optimization is only possible with a load rebalance.
We understand the concern. Our intention was to briefly explain the process, but maybe it was oversimplified. The idea is doing separate scalability analysis for each component. Then, a point in the scaling curve is chosen so that all the components can run efficiently (depending on the throughput/energy scenario, time to solution or energy to solution, the compromise will be different). Because of the different coupling/output frequencies of the components and because of eventual irregularities in the stepping, it is likely that the configuration has to be further tuned, by increasing the speed of one or other component, and ultimately looking at the load balance (examining the idle/waiting time of each one of the models).

We have modified the paragraph (L177-185 in revised manuscript) to highlight that 1) we are talking here about load rebalance (as the reviewer pointed out) once one of the components has been optimized and 2) the load rebalance is needed because there is a synchronization point at the end of each time-step where both components are waiting for fields from the other component.

Figure 2: change the label "SYPD" to "coupled EC-Earth3P-HR"?
Label has been changed.

L216: Where on Figure 3 can we see the 4 times of communication pattern?
Figure 3 has been modified to highlight the 4 communication patterns inside the coupling process including a new zoom, thank you. Taking into account the new addition, the text has been modified to reference the new zoom.

L225: I am not sure what parts on Figure 3 this paragraph refers to. Please clarify.
Figure 3 does not show this output process because the profiling events were not captured by the profiling tools (Extrae tool). The 30% of the time-step higher has been quantified from the execution time step. This has been clarified in the paragraph.

Figure 3: This figure is too noisy. Perhaps, the authors can replot it to better support the points they want to make with improved labels, organization, and clarity.
A new Figure 3 has been included, where each event is now more clearly distinguishable. Thank you for the advice. Some changes have been included into the text to be consistent with the new figure.

Figure 5: change "hist-1950" to italic
Done

L383: : : : in Table 2
Done

Figure 6: Add the global means and RMS errors (which give some overall ideas about the model performance) and discuss these numbers in the text. Change the title of figures to, for example, ": : :EC-Earth3-HR minus ERA-Int" to be clearer. Add labels (a) and (b). Also make these changes on other figures where applicable.
Thank you for your suggestions to improve this and other figures. We have computed the global means and RMS errors and discuss these numbers in the text. We have also changed the titles and added the labels.
In addition we have replaced ERA-interim with ERA5 because of the better quality of this new version of ERA (Hersbach et al 2020).

Hersbach, H., Bell, B., Berrisford, P., Hirahara, S., Horányi, A., Muñoz-Sabater, J., ... & Simmons, A. (2020). The ERA5 global reanalysis. *Quarterly Journal of the Royal Meteorological Society*

L408: change to ": : :Greenland (Fig. 9), which is : : : MSLP bias (Fig. 7a)."
Done.

L413: I would wonder whether enhancing horizontal resolution has a negative impact
on performance. The global mean biases and RMS errors (suggested above) are
helpful to provide some quantitative measure.
As discussed in the next sentence it can have a negative impact on the wet bias over the warm pool.

L430: Perhaps can add some figures to support this point.
The drift during the control run is shown in the figure below. It shows that the drift is very minor. The
largest drift of about 0.5 °C/100 year is in the 100-1000m layer. We have therefore decided not to
include this figure, but to add a sentence in the manuscript describing this minor drift more
quantitively.

[Figure]

Figure R1: Global mean ocean temperature of EC-Earth3P-HR averaged over depth for the control-
1950 simulation.

L444: Any explanations why this activation of deep convection at the Labrador Sea
occurs in the low-res version, but not in the high-res version?
Presently we have no clear understanding of this difference. It may be related to various aspects,
such as differences in meridional heat transport and differences in the resolution of sea-ice and
deep-convection, to mention a few. This is presently under investigation. A few lines to discuss this
are added in the text.

Figure 11: I suggest using different colors for different simulations, but similar ones
for the same resolution – redish for low-res; blackish for high-res. I also suggest the
authors add a panel of net radiation fluxes at the top of the atmosphere to show the
energy balance of the whole Earth system.
We have modified the figure according to the suggestions of the reviewer. We checked the net
radiation fluxes at the top of the atmosphere and they behave similar to the net surface heat fluxes,
which is to be expected due to the small heat capacity of the atmosphere. We therefore decided not
to include them in an extra panel.

Figure 13: Isn't it clearer to compare if the model results are shown in the same manner
(lines instead of bars) as the observation? I find it is difficult to follow the seasonal cycle
of EC-Earth3P – The base changes every month. Please revise it.

Thanks for this comment. We have modified the figure accordingly.

L534: trend -> drift? The trend on this line has a different meaning than the ones towards the end of the paragraph, so should use different words to distinguish.
We have changed "trend" into "drift" in the beginning of the paragraph to avoid confusion. Thanks.

L535: change to ": : :hist-1950 minus control-1950: : :"
Done

Figure 18: I understand the authors scale the right panel to fit the starting point of the curves. But it looks a bit weird to leave large white margins on it. Please revise.
We have revised the figure and removed the large white margins.

---

## Author Comment (AC2) · 11 Jun 2020

Response to Referee #2

In the present manuscript, the authors summarized basic model performance/drifts of EC-Earth3P-HR in comparison with lower resolution version, EC-Earth3P, together with optimization procedure of model code, data handling, how to post-process. The manuscript is well-organized, and basically I consider that the present manuscript will be worth publishing in GMD. In general, however, physical explanations on causes of model biases, drifts, and differences between EC-Earth3P-HR and EC-Earth3P are quite limited throughout the manuscript. After minor revisions in order to answer the suggestions and comments listed below, the manuscript will be more suitable for publication. We thank the reviewer for these positive remarks.

Comments
L. 373-374: How did you generate atmospheric temperature perturbations? Gaussian random noise with a certain amplitude? Please specify the method.
3D temperature perturbations are random samples from a uniform distribution over [-5e-5, +5e-5] degree. We have added this information in the manuscript.

L. 380: How did you change the oceanic mixing parameters? Please describe more details.
We added more details:

….It was therefore decided to change the ocean mixing parameters, which improved the AMOC. The main difference compared to the first ensemble member of EC-Earth3P is that the parameterization of the penetration of turbulent kinetic energy (TKE) below the mixed layer due to internal and inertial waves is switched off (nn_etau=0; Madec et al. 2016). The mixing below the mixed layer is an ad-hoc parameterization into the TKE scheme (Rodgers et al. 2014,) and is meant to account for observed processes that affect the density structure of the ocean's boundary layer. In EC-Earth3P, this penetration of TKE below the mixed layer caused a too deep surface layer of warm summer water masses in the North Atlantic convection areas which lead to a breakdown of the Labrador Sea convection within a few years and a strongly underestimated Atlantic Meridional Overturning Circulation (AMOC) in EC-Earth. An additional minor modification compared to ensemble member 1 is an increased tuning parameter rn_lc (=0.2) in the TKE turbulent closure scheme that directly relates to the vertical velocity profile of the Langmuir Cell circulation. Consequently the Langmuir Cell circulation is strengthened.

Rodgers, K. B., O. Aumont, S. E. Mikaloff Fletcher, Y. Plancherel, L. Bopp, C. de Boyer Montégut, D. Iudicone, R. F. Keeling, G. Madec, and R. Wanninkhof, 2014: Strong sensitivity of southern ocean carbon uptake and nutrient cycling to wind stirring. Biogeosciences, 11 (15), 4077–4098, doi:10. 5194/bg-11-4077-2014, URL HTTP://www.biogeosciences.net/11/4077/2014/.

Madec and the NEMO team 2016: NEMO ocean engine version 3.6 stable. Note du Pôle de modélisation de l'Institut Pierre-Simon Laplace No 27, ISSN No 1288-1619.

L. 399-408: There are almost no explanations on cause of model biases described here. Please give possible reasons or speculations for the biases which may be arisen from, for example, deficiencies in parameterizations for cloud microphysics, (deep,

shallow, strat) cumulus, insufficient horizontal resolution, albedo parameterization of snow, sea-ice, etc.

Without a detailed analyses of the origin of the biases it is difficult to know the causes. We have, however, included a discussion about the possible causes.

L. 411: Why the MSLP over Antarctica is higher (worse) in EC-Earth3P-HR than ECEarth3P? Please give possible reasons or speculations for the biases. In addition, if the biases in stationary eddies (Fig. 9) MSLP (Figs. 7 and 10) are evaluated, you may want to show wintertime storm track activity defined as subweekly eddy meridional temperature flux at the 850 hPa for EC-Earth3P-HR than EC-Earth3P, which may be useful for interpreting differences of model biases between two models.

We agree that this difference is remarkable and unexpected. More analyses is required to fully understand this. The difference is most strongly in the austral winter, which suggest that it is related to the dynamics of the polar vortex that is sensitive to the horizontal resolution. We have added a few lines in the text.

We agree that storm track activity is a useful diagnostic, but because this article focuses on basic validation this will be explored more in detail in future papers.

Figure 6, 7, and 8: In order to evaluate model errors quantitatively, please calculate root-mean-squared errors (RMSE) for EC-Earth3P-HR with respect to observations/ reanalysis and add the RMSE to somewhere in the corresponding figures, for example, just right of the figure title as "Diff DJF SAT EC-Earth3-HR ERA-Int (0.8 K)".

The global mean RMSE have been calculated and are mentioned in the figure captions, together with the global mean values.

Figure 10: Panels showing difference between EC-Earth3P-HR and EC-Earth3P may be replaced by the errors between EC-Earth3P and observations/reanalysis as in Figs. 6-8. And, RMSEs for EC-Earth3P may be given.

Because EC-Earth3P-HR and EC-Earth3P have similar error patterns compared to observations/reanalysis we have chosen to show the differences between EC-Earth3P and EC-Earth3P-HR to highlight the impact of resolution. The similarity in error between EC-Earth3P and EC-Earth3P-HR is also reflected in similar RMSE values. These are now mentioned in the figure captions and are also discussed in the text.

Figure 12: Top label "2040-2049 minus 1950-1959" may be wrong.

The label may indeed be confusing. We have removed it and expanded the figure caption to remove this confusion.

L. 444: Why does not the activation of deep convection in EC-Earth3P-HR occur and why do global-mean SAT and AMOC transport keep stable in EC-Earth3P-HR?

We do not know the cause yet. A few possible reasons are the impact of ocean resolution on the sea-ice dynamics and deep ocean convection. But also changes in the ocean temperature and salinity structure might play a role. This is presently under investigation. We have added a few lines in the text to discuss this.

Figure 14: For comparison, Figs. 14c and 14f may be replaced by observations. Also Z500 anomalies regressed onto NINO3.4 index, which can be superimposed onto Fig.

14de by contours, are useful for evaluating atmospheric teleconnection pattern.

Thanks for these two points. We have added both to Fig. 14, namely the analysis of observations (HadISST for SST and ERA-Interim for Z500) and the regression of Z500 onto Niño3.4 to report the ENSO teleconnection to the Northern Hemisphere. We have also modified the text accordingly.

L. 473-476: SST variability is closely related to the frontal structure seen in the climatic mean SST. So, you may want to add DJF climatic-mean SST to Figs. 14a-c by contours.

This is also a very relevant point; thanks. We have now included in Fig. 14-top the winter SST climatology for both, the model versions and HadISST.

Figure 15: Again, please add the corresponding panels for observations/reanalysis.

We have added the analysis of ERA-Interim to Fig. 15, and modified the text accordingly. Note that we have removed the citation of the observational paper in that passage.

L. 512: Please capitalize "Rapid".

Done.

L. 525: ERA-Interim is just reanalysis data, not observations. You may want to redraw the green lines in Fig. 18 based on observations, for example, HadCRUTv4.4 with keeping consistency of undefined grid point between observations and models. And then, please rewrite Section 4.2.4.

The undefined grid points of HadCRUTv4.4 at 1950 cover a large fraction of the globe as shown below. We argue that comparing the model over this limited region is not representative for the global warming of the model during the hist-1950 run. Although we recognize that ERA-interim is just reanalysis data it provides a reasonable estimate of the global warming. We have replaced the ERA-interim data by ERA5 in Fig. 18 for obtaining the most up to date reanalysis estimate.

[Figure]

Figure R2. Red: Grid points with T2m observations of HadCRUT4 for 1950.